# Polymorphic estrogen receptor binding site causes Cd2-dependent sex bias in the susceptibility to autoimmune diseases

Gonzalo Fernandez Lahore [1], Michael Förster[1], Martina Johannesson [1,6], Pierre Sabatier [2], Erik Lönnblom [1], Mike Aoun[1], Yibo He[1], Kutty Selva Nandakumar [1,3], Roman A. Zubarev [2,4] & Rikard Holmdahl [1,5✉]

Complex autoimmune diseases are sexually dimorphic. An interplay between predisposing genetics and sex-related factors probably controls the sex discrepancy in the immune response, but the underlying mechanisms are unclear. Here we positionally identify a polymorphic estrogen receptor binding site that regulates *Cd2* expression, leading to female-specific differences in T cell-dependent mouse models of autoimmunity. Female mice with reduced Cd2 expression have impaired autoreactive T cell responses. T cells lacking Cd2 costimulation upregulate inhibitory Lag-3. These findings help explain sexual dimorphism in human autoimmunity, as we find that *CD2* polymorphisms are associated with rheumatoid arthritis and 17-β-estradiol-regulation of CD2 is conserved in human T cells. Hormonal regulation of CD2 might have implications for CD2-targeted therapy, as anti-Cd2 treatment more potently affects T cells in female mice. These results demonstrate the relevance of sex-genotype interactions, providing strong evidence for CD2 as a sex-sensitive predisposing factor in autoimmunity.

[1] Division Medical Inflammation Research, Dept. Medical Biochemistry and Biophysics, Karolinska Institute, Solna, Sweden. [2] Division of Physiological Chemistry I, Department of Medical Biochemistry and Biophysics, Karolinska Institute, Solna, Sweden. [3] SMU-KI United Medical Inflammation Centre, School of Pharmaceutical Sciences, Southern Medical University, Guangzhou, China. [4] Department of Pharmacological & Technological Chemistry, I.M. Sechenov First Moscow State Medical University, Moscow 119146, Russia. [5] The Second Affiliated Hospital of Xi'an Jiaotong University (Xibei Hospital), 710004 Xi'an, China. [6] Present address: Division of Rheumatology, Department of Medicine Solna, Karolinska Institute, Karolinska University Hospital, SE-171 76 Stockholm, Sweden. ✉email: rikard.holmdahl@ki.se

Women generally mount a more vigorous immune response than men, and are more susceptible to most autoimmune diseases[1,2]. These diseases have a strong but complex genetic component, and it has been difficult to identify the underlying polymorphisms[3–5]. The female preponderance in autoimmunity is controlled by sex hormones[6] and genetics[7], not only through sex chromosomes but also through sex hormone-mediated regulation of autosomal genes. However, conclusive evidence is lacking, as it is difficult to positionally identify the underlying polymorphisms controlling complex traits in a sex-dependent manner.

Analysis of genetically segregated inbred animal strains dramatically enhances the power to isolate polymorphisms underlying complex diseases. Compared with association studies of human cohorts, studies in mice reduce environmental variability and allow for proof-of-concept experiments in biologically relevant systems, making it possible to conclusively identify genes underlying complex traits. In the context of previous such work to identify genetic loci that regulate autoimmune arthritis[8–10], our research group identified a locus on mouse chromosome 3 (Cia21) that affects expression of the T cell activation marker Cd2 and regulates arthritis severity in females, but not in males[9].

Here we report the causative mechanism to be a single polymorphism in an oestrogen receptor binding site (ERBS) within Cia21. This polymorphic ERBS orchestrates the expression of its surrounding genes in a sex-specific manner, including Cd2. We isolate this polymorphic ERBS in a congenic mouse line (D3-31) and use these mice to study the consequences of oestrogen-mediated regulation of Cd2 for T cell-dependent autoimmunity. Importantly, we find oestrogen regulation of CD2 expression to be a conserved mechanism in humans likely contributing to the sexual dimorphism in T cell-mediated autoimmune diseases.

## Results

We have set out to identify major genetic polymorphisms underlying the development of autoimmune arthritis, using animal models. As part of these efforts a major quantitative trait locus (QTL) was identified on chromosome 3 qF2.2, which was termed Cia21[9]. Cia21 was identified from an intercross between the collagen-induced arthritis (CIA)-susceptible C57BL/10.RIII (BR) and the CIA-resistant RIIIS/J (R3) mouse strains[11]. Cia21 contains several differentially expressed genes, including Cd2 and Ptpn22[9]. Both Cd2 and Ptpn22 play a key role in T cell activation and were proposed as strong candidate genes. The aim of the present study is to identify the polymorphisms underlying the Cia21 QTL.

**A minimal non-coding genetic interval proximal to Cd2 explains Cia21**. To dissect the Cia21 QTL, we bred heterozygous Cia21 mice and evaluated the resulting recombinant mice (shown in Fig. 1a) using CIA (Fig. 1b–g). Out of all the evaluated recombinants, only two, numbers 1 and 5, recapitulated the protective arthritis phenotype previously observed in Cia21 mice[9]. Thus, the Cia21 QTL results from individual contributions of these two sub-QTLs. Importantly, the phenotype driving recombinant regions 1 and 5 mapped to the previously proposed[9] candidate genes Cd2 and Ptpn22, respectively. Recombinant fragment 1 (proximal to Cd2), however, was significantly smaller than fragment 5 providing better conditions for the positional identification of underlying polymorphisms. Therefore, we focused our efforts on the former.

Recombinant fragment 1 stretched from markers D3KV1 to MF31 (Fig. 1a, ca. 0.2 Mbp), but could be further redefined to the significantly smaller D3KV1-MF96 interval (ca. 0.02 Mbp) through a recombination assisted breeding strategy. Although

recombinant fragments 1, 2 and 3 overlapped significantly, only fragment 1 regulated arthritis. Thus, we concluded that the causative polymorphisms must be positioned between markers D3KV1 and MF96 (Fig. 1a, highlighted yellow). D3KV1-MF96 is a non-coding 0.02 Mbp region proximal to Cd2, located in-between the genes Atp1a1 and Igsf3 (Fig. 1h). We isolated the D3KV1-MF31 recombinant fragment (termed D3-31) in a congenic mouse line for further investigations. D3-31 congenic mice carry the parental R3 allele of D3-31 on an otherwise BR background. For simplicity, we hereon refer to the congenic line as D3-31 and to wild-type littermates as BR.

**Female D3-31 mice are protected from T-cell-dependent models of autoimmunity**. In accordance with previous data on Cia21, the R3 allele of D3-31 protected congenic mice in T-cell-dependent[12–14] autoimmune inflammatory models, including collagen-induced arthritis (CIA), experimental autoimmune encephalomyelitis (EAE) and delayed type hypersensitivity (DTH) (Fig. 2a–f). We also investigated the T-cell-independent[15] collagen antibody-induced arthritis (CAIA) model, but observed no phenotypic differences (Supplementary fig. 1). As the DTH model does not depend on B cells[12], these results indicated a critical role for T cells. Interestingly, and as previously described for Cia21[9], only female D3-31 mice were protected from T-cell-mediated autoimmunity (Fig. 2a–f). Thus, we concluded that D3-31 regulates T-cell-dependent autoimmune phenotypes, and likely T cells, in a sex-specific manner.

**Female sex hormones are required for the protective phenotype in D3-31 mice**. To discriminate between influence of sex chromosomes versus hormones, we performed CIA and EAE experiments in castrated female mice (Fig. 3a–c). Castration of female mice depletes gonadal production of 17-β-estradiol (E2)[16], which constitutes the major circulating oestrogenic compound in females. Castration reverted the protective effect of the D3-31 fragment both in CIA and EAE (Fig. 3a–c), which demonstrated the crucial contribution of female sex hormones, most likely E2, to the protective phenotype in female D3-31 mice. We next set out to define the genetic mechanisms underlying this sexually dimorphic immune phenotype by sequencing the D3-31 fragment.

**Polymorphic ERBS in D3-31 interval affects E2-mediated transcriptional activity**. DNA sequencing of the D3-31 BR and R3 alleles revealed four single-nucleotide polymorphisms (SNPs) in the critical D3KV1-MF96 interval (Fig. 4a, b). None of the variants affected the coding region of known genes, indicating distal (cis) regulation of gene expression, likely by interfering with regulatory elements. Given our previous observations, we speculated that the identified polymorphisms could be located within an ERBS, interfering with sex-dependent regulation of gene expression.

Oestrogen receptors (ERα and ERβ) are nuclear hormone receptors that translate E2-mediated signalling. Both ERα and ERβ are expressed in immune cells[17], and act as transcription factors regulating the expression of proximal and distant genes[18,19]. To test our hypothesis, we screened publicly available ChIP-seq data for Erα binding sites overlapping with one or more of the sequenced SNPs within D3KV1-MF96 interval. Indeed, one of the SNPs, AC > GG on chr3:101310478-479 (termed SNP478), clearly overlapped with an Erα binding site (Fig. 4c). In fact, bioinformatic analysis also revealed an oestrogen response element (i.e. an ER core binding motif) in close proximity to SNP478. We sought to verify this finding and could confirm binding of Erα to SNP478 in mouse spleen cells using ChIP-qPCR (Fig. 4d). Comparison of

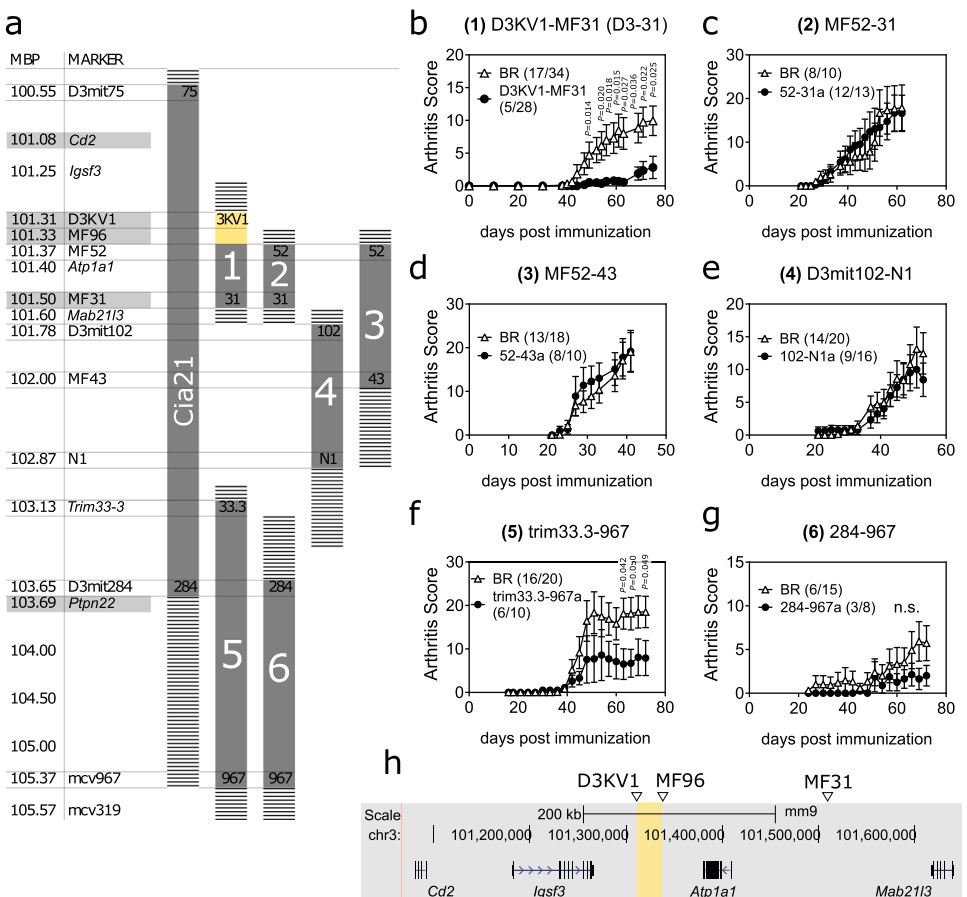

**Fig. 1 Map of Cia21 QTL and critical D3KV1-MF31 interval.** The Cia21 QTL resulted from an intercross between the CIA susceptible C57BL/10.RIII (BR) and the CIA-resistant RIIIS/J (R3) strains. Cia21 is present on chromosome 3 qF2.2 and is 3 Mbp in size. **a** Schematic representation of the Cia21 QTL and recombinant mice derived by intercrossing of Cia21 heterozygotes. Important genetic markers and genes are indicated on the left. The critical D3KV1-MF96 interval is highlighted in yellow. Uncertainty borders are dashed. **b–g** Collagen-induced arthritis in female recombinant mice from (**a**) compared to BR littermate controls. Incidence and total number of mice are indicated in parenthesis on the respective graphs. Data are summarised as mean (SEM). Statistical significance was evaluated using a two-tailed non-parametric Mann–Whitney U test. Data in each graph was pooled from two independent experiments. **h** Detailed view of D3KV1-MF31 (fragment 1, D3-31) and proximal genes. The critical D3KV1-MF96 interval is highlighted in yellow. Coordinates according to mouse NCBI37/mm9 build. n.s. not significant.

SNP478 between mouse inbred strains revealed that this SNP is in fact part of a highly polymorphic AC/GT simple repeat (Supplementary fig. 2, extracted from Kent et al.[20]).

To address whether SNP478 had functional consequences for E2-mediated transcriptional activity (i.e. interfered with the binding of Erα to the DNA), we cloned the candidate D3KV1 ERBS (±100 bp) in its two variant forms (AC and GG) into luciferase reporter constructs. Leveraging the fact that human and mouse ERα are highly conserved[21], we assessed transcriptional activity of these constructs in the ERα expressing MCF-7 human cell line, treating them with increasing concentrations of E2 (Fig. 4e). In the context of the reporter construct, an increased occupancy of the candidate ERBS by ERα (as a function of increasing E2) resulted in dose-dependent suppression of transcriptional activity. Although counterintuitive, similar observations have been reported elsewhere[22]. Given the stronger transcriptional inhibition when using the BR derived construct, we concluded that ERα/Erα has a higher affinity for the BR allele than for the D3-31 allele. Importantly, these data demonstrate that SNP478 has functional consequences for E2-mediated transcriptional activity.

**Polymorphic ERBS in D3-31 interval leads to female-specific changes in *Cd2* expression.** Next, we sought to test the biological

relevance of our findings by comparing the gene expression profile in lymph node cells from male and female D3-31 and BR mice. We observed female-specific changes in the expression of three genes adjacent to the polymorphic ERBS, namely *Cd2*, *Igsf3* and *Mab21l3* (Fig. 5a, b). We also investigated the expression of *Atp1a1* as well as more distal genes (*Cd101* and *Slc22a15*) previously implicated in the non-obese diabetic (NOD) mouse model of type 1 diabetes[23], but found no changes in their expression level. Notably, the female-specific reduction of Cd2 expression in D3-31 mice was also evident at protein level (Fig. 5c), correlating with our gene expression results and those previously reported in Johanesson et al.[9].

Out of the differentially expressed genes, *Cd2* was the only gene predominantly expressed in lymphoid tissue (Fig. 5d), particularly in activated Cd4[+] T cells (Fig. 5e). *Igsf3* and *Mab21l3* regulate neural[24] and ocular[25] development, whereas Cd2 has been involved in immune function[26] and associated with human autoimmune conditions[4,27]. Indeed, treatment of lymph node cells with anti-Cd2 mAb inhibited T cell activation as demonstrated by reduced secretion of pro-inflammatory cytokines (Fig. 5f). Considering these data and normal development of D3-31 mice, we concluded that Cd2 is driving the T-cell-dependent immune phenotype observed in D3-31 mice.

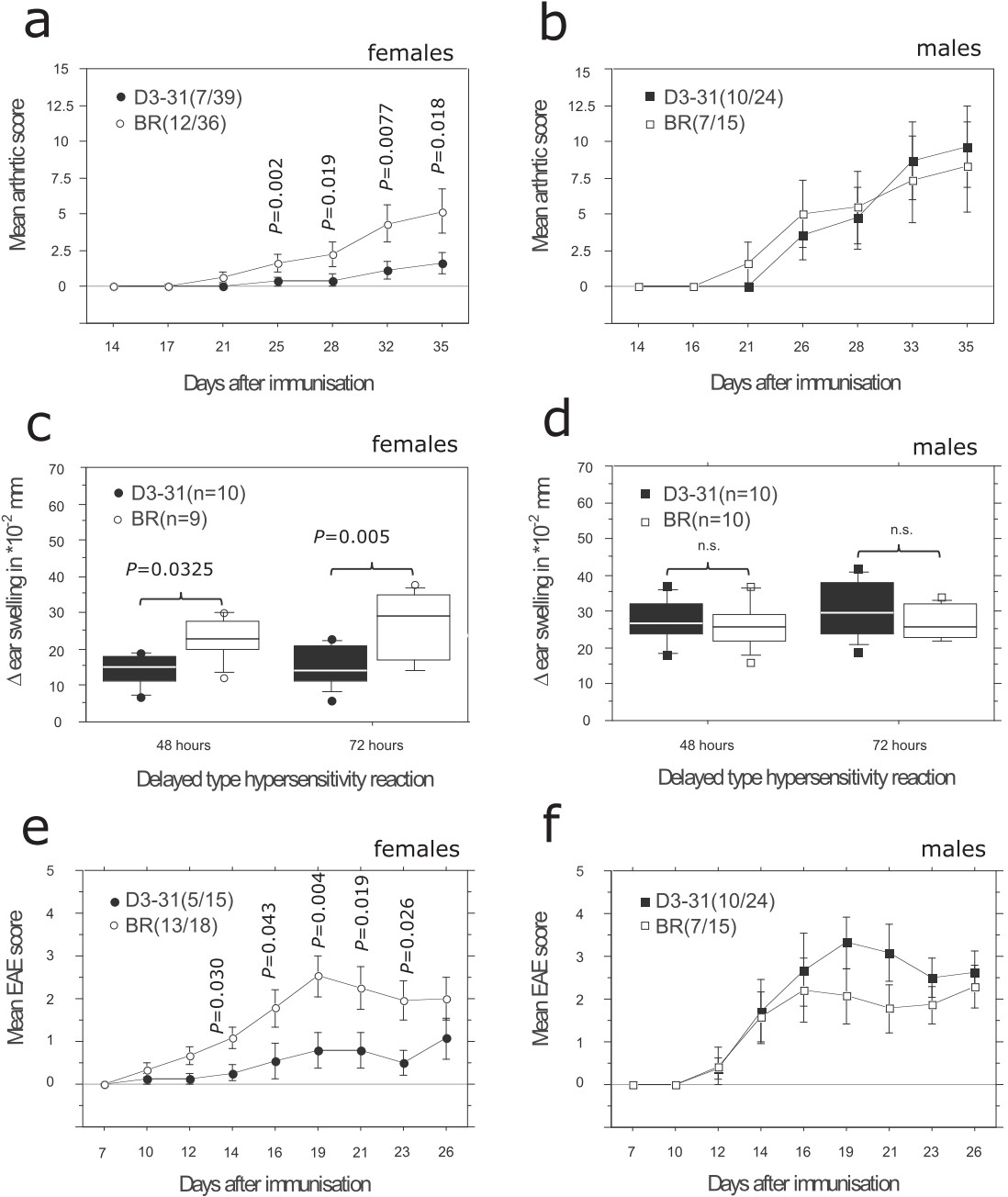

**Fig. 2 D3-31 mice are protected from T cell-dependent autoimmunity in a sex-specific manner.** Collagen-induced arthritis (CIA) in **a** female and **b** male BR and D3-31 mice. Data was pooled from two independent experiments. Delayed-type hypersensitivity (DTH) reaction in **c** female and **d** male BR and D3-31 mice. Box upper and lower limits indicate interquartile range (25th/75th percentiles), the middle line indicates the median. Whiskers indicate 10th and 90th percentiles. Min. and max. values are plotted as individual dots. Data is representative of three independent experiments with similar results. MBP89-101-induced experimental autoimmune encephalomyelitis (EAE) in **e** female and **f** male BR and D3-31 mice. Data was pooled from two independent experiments. For all graphs, incidence and total number of mice used are indicated in parenthesis. Data are summarised as mean (SEM). Statistical significance was evaluated using a two-tailed non-parametric Mann–Whitney $U$ test.

Given the sex-specific differences in gene expression, we next investigated the relation between E2 and Cd2 expression. T cells cultured in the presence of E2 upregulated Cd2 in a dose-dependent manner (Fig. 5g). Conversely, use of E2 depleted medium (achieved by using charcoal-stripped serum) reduced the expression of Cd2, and, more importantly, neutralised the observed differences in Cd2 expression between BR and D3-31 mice. Additionally, differences in Cd2 expression could be re-established by reintroducing E2 to the medium (Fig. 5h). This not only demonstrates direct regulation of E2 on Cd2 expression, but

also proves that the identified polymorphisms interfere with this regulation. Consequently, we speculated that E2-mediated regulation of Cd2 was contributing to sex-specific differences in the T cell responses. A sex-dependent reduction of Cd2 expression in female D3-31 mice could likely limit the T cell responses.

**E2-dependent regulation of Cd2 leads to sex-specific differences in T cell activation.** To investigate the impact of sex hormone-dependent alterations in Cd2 expression on the T cell

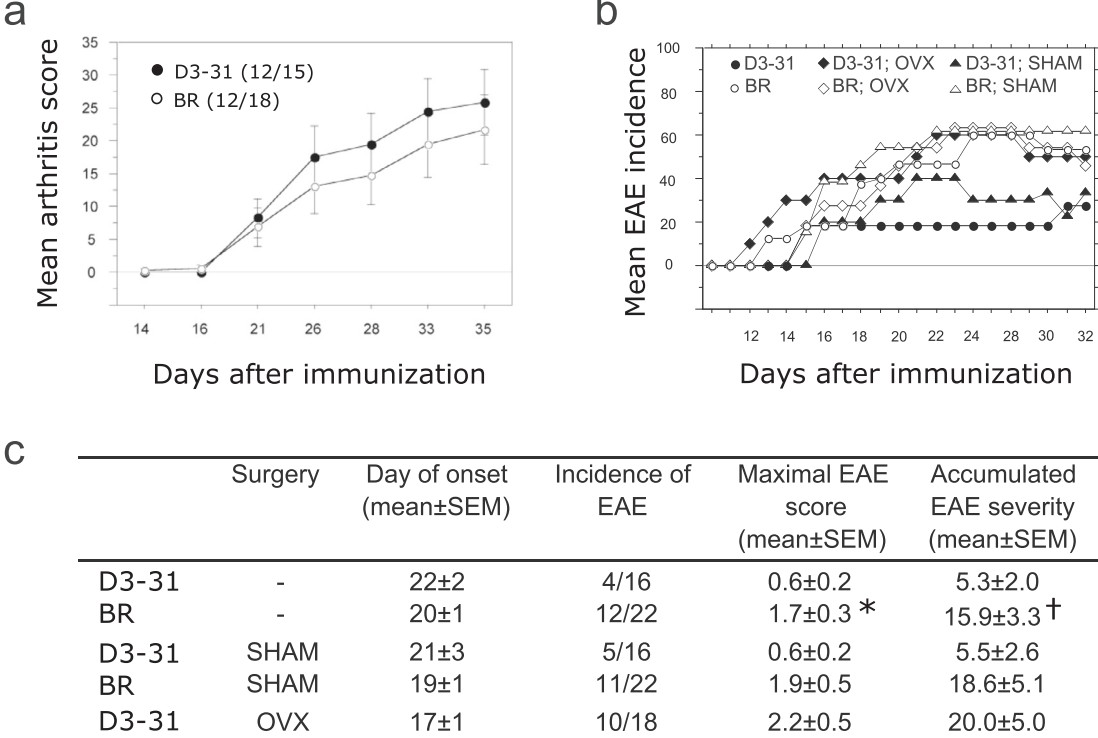

**Fig. 3 Female sex hormones are required for the protective phenotype in D3-31 mice. a** CIA severity and incidence (in parenthesis) in ovariectomized D3-31 and BR mice. Data was pooled from two independent experiments. **b** Incidence of EAE in ovariectomized (OVX) and sham operated (SHAM) D3-31 and BR mice. **c** Table comparing incidence, maximal score and accumulated severity of the EAE experiment shown in **b**. Data was pooled from three independent experiments. Data are summarised as mean (SEM). Statistical significance was evaluated using a two-tailed non-parametric Mann–Whitney $U$ test. *$P = 0.0486$; †$P = 0.0419$.

responses, we compared the activation of T cells between BR and D3-31 female mice. In a first set of in vitro experiments, we found an impaired response in D3-31 T cells to Tcr stimulation, as evidenced by reduced proliferation and Il-2 production (Fig. 6a, b). Importantly, the difference in T cell proliferation between BR and D3-31 mice could be enhanced in a dose-dependent manner by E2 (Fig. 6c), much like the E2-dependent expression differences observed for Cd2 (Fig. 5h).

A diminished T cell response in D3-31 mice was also evident in vivo. Compared to BR mice, D3-31 mice showed a lower level of antigen-specific T cells responses 10 days after immunisation with CIA antigen bovine collagen type II, as demonstrated by reduced secretion of pro-inflammatory cytokines in lymph node cell cultures recalled with antigen (Fig. 6d). Flow cytometry analysis of D3-31 draining lymph node cells revealed lower numbers of antigen experienced $Cd40l^+$ $Cd4^+$ T cells (Fig. 6e), which expressed reduced levels of Cd2 (Fig. 6f) and Il-17a after ex vivo restimulation with PMA (Fig. 6g, h). Differences in T cell activation status were also evident given lower numbers of induced regulatory T cells after immunisation (Fig. 6i, j). Importantly, the observed differences in T cell activation were strictly sex-specific (Fig. 6e–j), mirroring sex-specific differences in Cd2 expression. Treatment with anti-Cd2 strongly reduced the expression of Il-17a in autoreactive $Cd4^+$ T cells (Fig. 6h), as well as expression of Il-17a and Foxp3 in naïve T cells (Fig. 6k), demonstrating the importance of Cd2 costimulation for the differentiation of Th17 and Treg type cells. Consequently, we concluded that reduced Cd2 expression in female D3-31 mice limits T cell activation in a sex-specific manner.

To further characterise how impaired Cd2 signalling affected T cells, we compared the proteomic landscape of anti-Cd3-stimulated $Cd4^+$ T cells in the presence or absence of anti-Cd2 mAb. Blocking of Cd2 signalling by means of anti-Cd2 mAb resulted in the selective upregulation of the immune inhibitory marker Lag-3 (Fig. 6l–n). Thus, we concluded that Cd2 costimulation is required for T cell activation and that impaired Cd2 signalling results in the upregulation of the inhibitory marker Lag-3.

**CD2 is associated with rheumatoid arthritis and is regulated by E2 in humans.** Our results in mice suggested a regulatory role for *CD2* on T-cell-dependent autoimmunity, which is genetically determined in a sex-linked manner. We therefore went on to explore the relevance of our findings in humans, specifically in the context of rheumatoid arthritis (RA). In a genetic association study, we found a significant association between *CD2* polymorphisms and RA (Fig. 7a and Supplementary fig. 8). While this association was more often found in females than in males, this was likely due to higher prevalence of RA in females (female to male ratio 3:1). Interestingly, several of the SNPs associated with RA can enhance expression of *CD2* (Fig. 7b), as we determined from the GTEx database[28]. Further analysis of available microarray datasets[29] revealed a mild yet significant correlation between *CD2* expression in RA synovia and disease activity (Fig. 7c). Moreover, *CD2* is strongly upregulated in the synovial tissue from RA patients when compared to osteoarthritis or healthy synovium (Fig. 7d). Thus, it is likely that CD2 is involved in joint inflammation, and that *CD2* polymorphisms affecting its expression contribute to the development or perpetuation of joint autoimmunity.

Importantly, women expressed higher levels of *CD2* than men, both in RA synovium and healthy PBMCs (Fig. 7c, e,

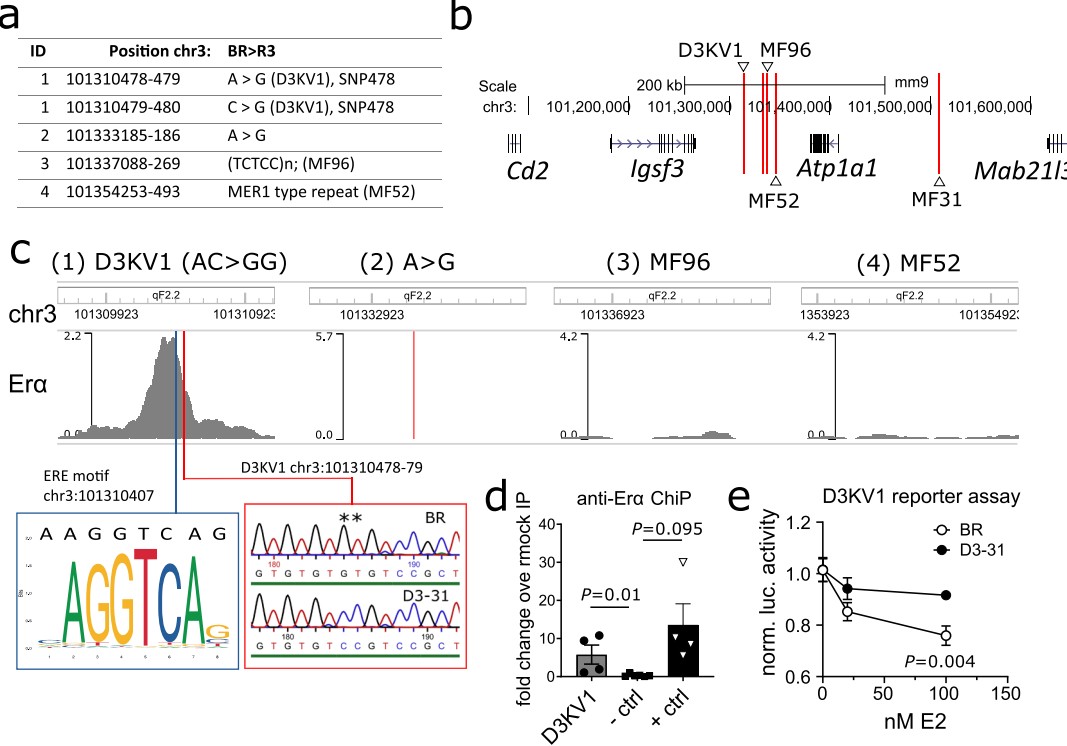

**Fig. 4 Polymorphism in D3-31 ERBS affects E2-mediated transcriptional activity. a** Sequencing results showing genetic variants within critical D3KV1-MF96 interval. **b** Detailed schematic overview of polymorphisms (denoted by red lines) in the D3KV1-MF96 interval. SNP478 denotes an AC > GG substitution on chr3:101310478-79. **c** ChIP-seq data from mouse uterus (extracted from SRX129062[63]) showing Erα binding intensity to polymorphic regions listed in **a**. Consensus ER binding motif (UN0308.1[64]) and SNP478 are highlighted in blue and red squares, respectively, where double asterisk indicates the position of SNP478. Coordinates according to mouse NCBI37/mm9 build. **d** Rabbit anti-mouse Erα ChIP-qPCR data confirming binding of Erα to SNP478 in spleen cells. A gene dessert was used as negative control (-ctrl) and a known Erα binding site (*Csf2ra*[30]) as positive control (+ ctrl). Values are expressed as fold enrichment over rabbit IgG mock IP. Each dot represents an independent mouse biological replicate. The data shown is representative of two independent experiments with similar results. **e** Effect of SNP478 on the transcriptional activity of the D3KV1 Erα binding site shown in **c**. The candidate D3KV1 Erα binding site (chr3:101310478 ± 100 bp to each side) was cloned in its two variant forms (AC and GG) into luciferase reporter constructs. The constructs were transfected into MCF7 cells to evaluate transcriptional activity. The data shown is from a total *n* = 9 technical replicates pooled from two independent experiments. Data are summarised as mean (SEM). Statistical significance was evaluated using a two-sided non-parametric Mann–Whitney *U* test.

respectively), suggesting the E2-mediated regulation of *CD2* observed in mice is conserved in humans as well. To corroborate our findings, we stimulated CD4+ T cells from healthy human donors with increasing amounts of E2. Firstly, we noticed a strong upregulation of CD2 in antigen experienced CD45RO+ T cells compared to their naïve CD45RA+ counterparts (Fig. 7f, g). But more importantly, expression of CD2 could be enhanced in CD45RO+ T cells by incubation with E2 in a concentration-dependent manner (Fig. 7h). Indeed, analysis of available ChIP-seq data[30] revealed that ERα robustly binds the human *CD2* gene locus (Supplementary fig. 8). Thus, these data demonstrate the evolutionary conserved nature of E2-mediated regulation of CD2.

**Anti-Cd2 mAb treatment affects T cells in female mice more than in male mice.** We reasoned that hormonal regulation of CD2 expression could have implications for anti-CD2-mediated therapy, as previous research suggests that anti-CD2 (Alefacept) preferentially targets CD2hi T cells[31]. To test this, we compared the in vivo effects of anti-Cd2 mAb administration on circulating T cells from male and female mice (Fig. 8). Anti-Cd2 mAb treatment partially depleted circulating T cells and resulted in a relative expansion of effector Cd44+ T cells in the remaining T cell pool, skewing the naïve Cd62L+/effector Cd44+ T cell ratio (Fig. 8c–h). This effect was significantly more pronounced in females, which, like in humans, expressed higher levels of Cd2 in

circulating T cells. Taken together, these data demonstrate that sex-dependent differences in Cd2 expression determine the response to anti-Cd2 mAb.

## Discussion
Using forward genetics, we have positionally identified a polymorphic ERBS regulating T-cell-dependent autoimmunity. This site orchestrates expression of surrounding genes in a sex-specific manner, including expression of the T cell co-stimulatory molecule, Cd2. We find that E2-mediated regulation of Cd2 is a conserved mechanism that influences T cell activation in a sex-specific manner, contributing to the sexual dimorphism in autoimmune diseases.

Understanding the sexual dimorphic immune responses is fundamental for personalised medicine but is methodologically challenging. Common approaches to study this phenomenon rely on intricate manipulation of gonadal or hormonal systems[32], which has yielded valuable insights but with limited physiological relevance. Our study provides a more physiological perspective by the identification of a naturally occurring polymorphism in an ERBS, which enables studies on sex-associated differences in T-cell-mediated autoimmunity. Since we used a hypothesis-free approach, our findings strongly suggest E2-mediated regulation of CD2 as a key physiological mechanism contributing to sex

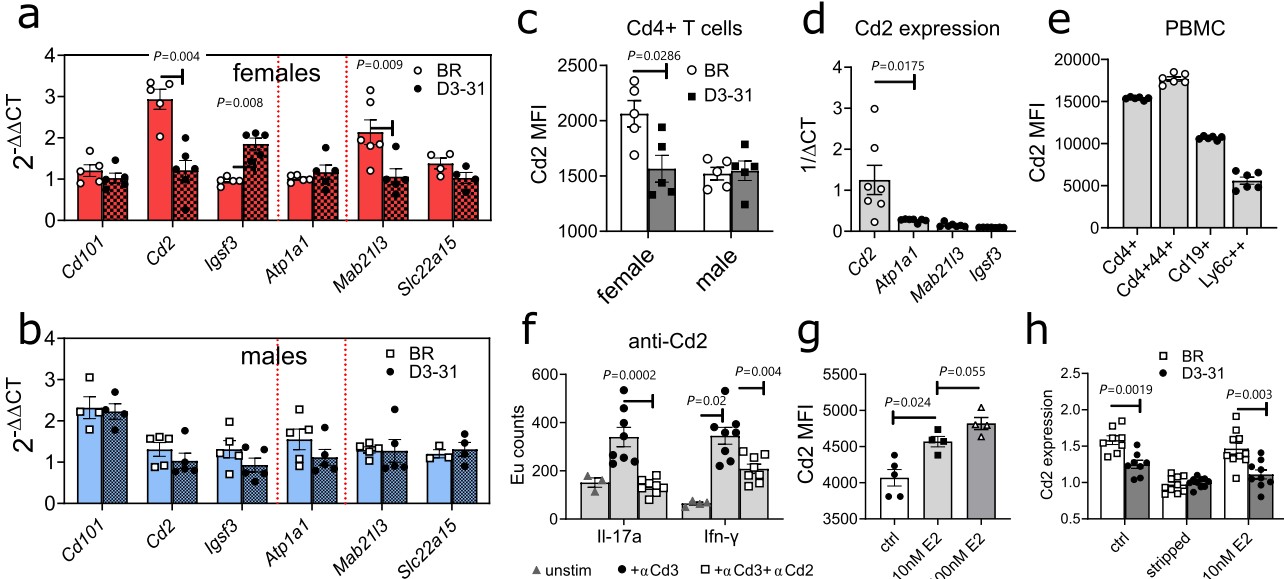

**Fig. 5 D3-31 mice show sex-specific differences in Cd2 expression. a, b** Expression of genes surrounding the D3-31 congenic fragment in lymph nodes cells. Expression data for female mice is shown in **a** (red), and for male mice in **b** (blue). Dotted red lines indicate congenic fragment borders. **c** Cd2 protein expression in lymph node Cd4+ T cells from female and male D3-31 and BR mice (flow cytometry). Data in **a**–**c** is representative of three independent experiments with similar results. **d** Expression of *Cd2* and other surrounding genes in lymph node cells from BR mice. **e** Cd2 protein expression in blood T cells, B cells and monocytes (flow cytometry). **f** Secretion of Il-17a and Ifn-γ in T cells stimulated with soluble anti-Cd3 mAb only, or soluble anti-Cd3 and soluble anti-Cd2 mAb. **g** Cd2 expression in lymph node T cells after in vitro culturing with increasing concentrations of 17-β-estradiol (E2). Data in **d**–**g** is representative of two independent experiments with similar results. **h** Comparison of Cd2 expression in T cells from D3-31 and BR mice cultured in normal medium (ctrl), charcoal-stripped medium devoid of E2 (-E2), or -E2 medium supplemented with 10 nM E2. Data in **h** is pooled from two independent experiments. In all figures, each dot represents one independent mouse biological replicate. Data are summarised as mean (SEM). Statistical significance was evaluated using a two-tailed non-parametric Mann–Whitney U test. Sequential flow cytometry gating strategies for **c** and **e** are provided in Supplementary fig. 9.

differences in T cell responses and the susceptibility to autoimmunity.

Our results also highlight the importance of genotype-sex interactions for the sexual dimorphism in autoimmune diseases. While much attention has been devoted to the contribution of sex chromosomes, epigenetic mechanisms or direct actions of hormones to the sexually dimorphic immune responses, the interactions between sex and predisposing autosomal polymorphisms have remained elusive. Isolated studies have demonstrated sex-dependency of expression (e) QTLs[28] and sex differences in the genetic associations to inflammatory diseases[33–35], but evidence is limited. Using a hypothesis-free approach, we conclusively identify a sex biased QTL with direct consequences for the development of autoimmunity. Polymorphisms in an ERBS within this QTL modulate E2-driven Cd2 expression, leading to sex-specific differences in T cell autoimmunity. Our results demonstrate not only that genetic polymorphisms influence hormonal regulation of gene expression, but also that genotype-sex interactions shape the sexually dimorphic immune response.

Independent of our sex-related findings, this study provides valuable insights into CD2 immunobiology. Polymorphisms in the *CD2* locus have been previously associated with several autoimmune diseases[4,27], but not much attention was given to the mechanism of action of these polymorphisms. Similarly, CD2 has been explored as a therapeutic target, but its mechanism of action beyond depletion of circulating T cell is poorly characterised[36], and complex adverse effects (including malignancies[37]) warrant further research. CD2 in isolation affects the formation of the immune synapse[38,39] and T cell activation[40,41], but the relevance of these findings in vivo are less clear. For example, targeting Cd2 in mice does not seem to affect immune system development[42] or thymic T cells[43], unless Tcr transgenic systems are used[44,45].

Thus, there is a need to study this therapeutically promising pathway in a physiologically relevant context. The D3-31 mice used in this study exhibit discrete changes in Cd2 expression mediated by E2, thus enabling us to study the effect of Cd2 on T-cell-mediated autoimmunity in a physiological setting.

We show for the first time that changes in Cd2 expression, caused by natural polymorphisms, affect the T cell responses. Reduced Cd2 expression protected mice from T-cell-dependent inflammation and autoimmunity by reducing the activation and proliferation of antigen-specific T cells. This is consistent with studies demonstrating that CD2 membrane density is proportional to TCR signalling strength[38], and that peptide-based blocking of Cd2 signalling reduces CIA severity[46]. Our results also implicate CD2 in the generation of Th17 and Treg-type responses. Mice with reduced Cd2 expression had a diminished T cell response characterised by a reduced expansion of Th17 and Treg cells. Accordingly, blocking Cd2 resulted in the suppression of both cell types in vitro. Indeed, CD2 has been linked to Treg[47,48] and Th17 phenotypes[39] before, and targeting CD2 is effective in the treatment of Th17-mediated inflammatory diseases like psoriatic arthritis[49]. In summary, these data suggest a key role for CD2-mediated activation in the induction of Th17 and Treg cells.

Mechanistically, CD2 seems to play a role in T cell activation beyond its ability to stabilise the immune synapse, as blocking Cd2 resulted in selective upregulation of the exhaustion marker, Lag-3. This finding is supported by studies showing an inverse correlation between CD2 expression and T cell exhaustion[38,50], and others showing upregulation of LAG-3 in CD8+ T cells after treatment with anti-CD2[51]. Taken together, the data suggest that costimulation through CD2 is important for proper T cell activation, and that impaired CD2 signalling results in upregulation of inhibitory or anergy markers such as LAG-3.

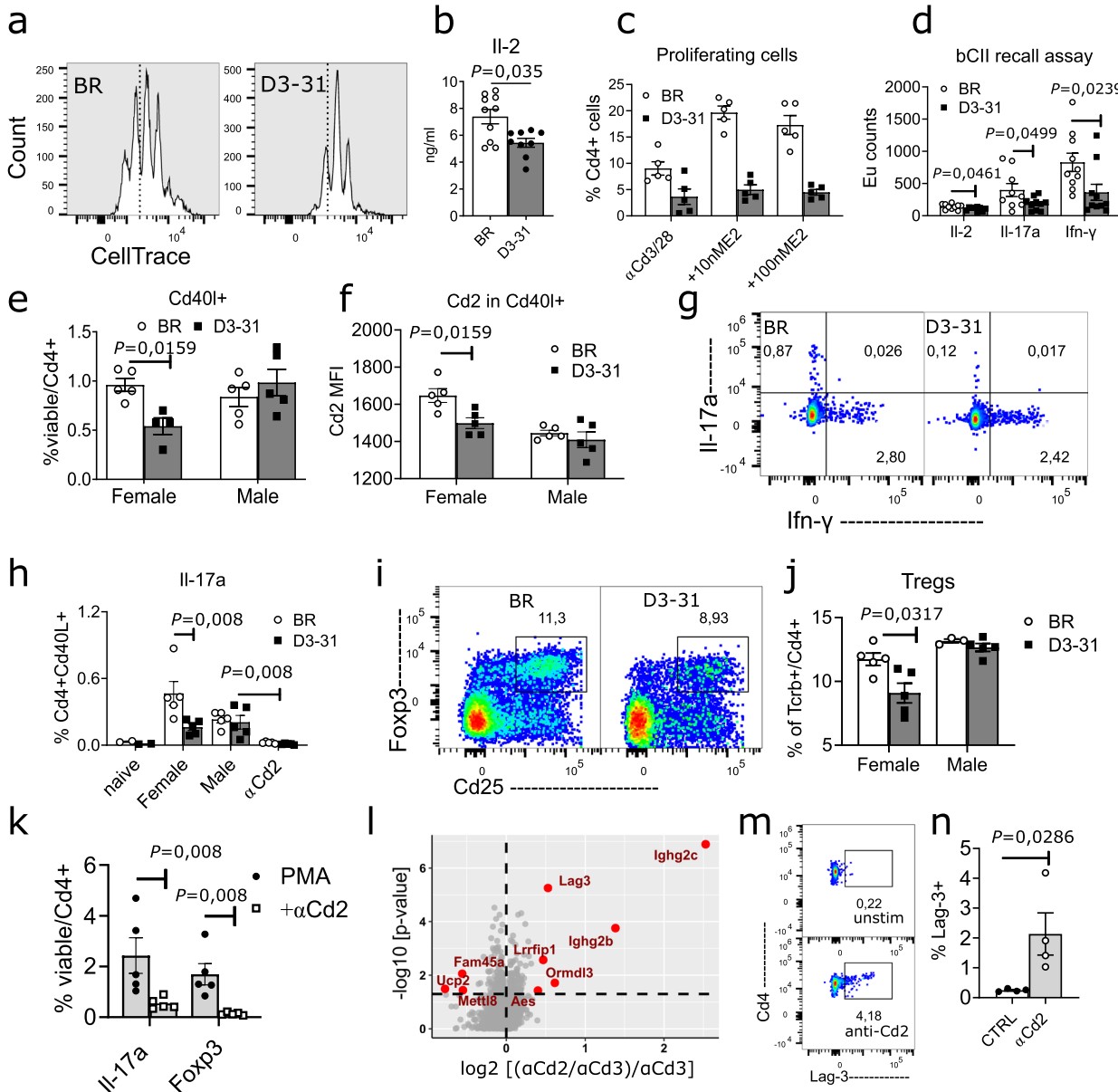

**Fig. 6 Sex-specific differences in Cd2 expression limit the T cell responses in female D3-31 mice. a**, **b** Proliferation (**a**) and Il-2 secretion (**b**) of Cd4+ lymph node T cells after stimulation with anti-Cd3/anti-Cd28 mAbs. **c** Proliferation of BR and D3-31 Cd4+ T cells as in **a** in the presence of increasing concentrations of E2 (10–100 nM). **d** Antigen recall assay showing pro-inflammatory cytokine secretion by lymph node cell cultures from CIA mice after recall with bovine collagen type II (bCII). Lymph nodes were harvested 10 days after immunisation with bCII (day 10). **e**, **f** Quantification of antigen experienced Cd40l+Cd4+ T cells in lymph nodes from CIA mice (day 10) (**e**), and expression of Cd2 in these cells (**f**). **g**, **h** Representative gating (**g**) and quantification (**h**) of Il-17a+Cd40l+ lymph node T cells from CIA mice (day 10) after ex vivo restimulation with PMA in the presence or absence of soluble anti-Cd2 mAb. **i**, **j** Representative gating (**i**) and quantification (**j**) of Cd25+Foxp3+ Tregs in lymph nodes from CIA mice (day 10). **k** Expression of Il-17a and Foxp3 in Cd4+ naïve T cells stimulated with PMA in the absence or presence of anti-Cd2 mAb. The data in **a–k** are representative of two independent experiments with similar results. **l** Volcano plot comparing the proteomic profile of Cd4+ T cells stimulated with immobilised anti-Cd3 mAb in the presence and absence of immobilised anti-Cd2 mAb. This experiment was performed once with n = 8 independent mouse biological replicates per group. **m**, **n** Flow cytometry data showing Lag-3 expression in Cd4+ T cells after culture with anti-Cd2 mAb. Representative gating in **m** and quantification in **n**. This experiment was performed three independent times with similar results. In all figures, each dot represents one independent mouse biological replicate. Data are summarised as mean (SEM). Statistical significance was evaluated using a parametric two-tailed t-test in **d**, and non-parametric two-tailed Mann–Whitney U test in all other experiments. Sequential flow cytometry gating strategies for **a**, **e–j**, **m**, **n**, as well as cell purity for **l** are provided in Supplementary figs. 10–15.

Our findings in mice are likely relevant to the sexual dimorphism observed in human autoimmune conditions. *CD2* associates with RA and E2 regulation of CD2 expression is highly conserved in human T cells. Women, who are generally more prone to autoimmunity, express higher levels of *CD2* than men. In mice, we demonstrate that these type of discrete

and sex-specific differences in Cd2 expression result in sexually dimorphic T cell responses that affect autoimmune phenotypes. Thus, subtle, physiological changes in CD2 expression caused by natural polymorphisms likely modify the risk of T cell-dependent autoimmunity in humans. E2-mediated regulation of CD2 probably contributes to sex differences in the

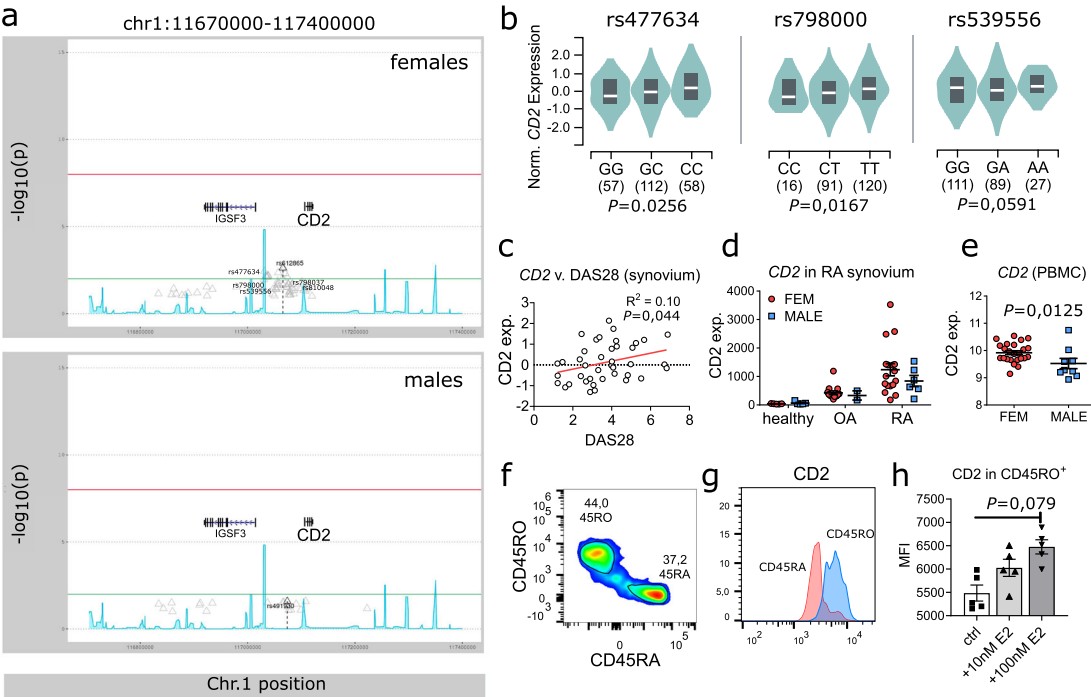

**Fig. 7 E2-mediated regulation of CD2 is conserved in humans. a** Genetic association data showing association between *CD2* polymorphisms and rheumatoid arthritis (RA) in female (top) and male (bottom) patients (EIRA cohort, *n* = 1341 males and 3361 females). **b** Effect of indicated SNPs on expression of *CD2* in human spleen as determined from GTEx database[65]. Number of samples is given in parenthesis below the graphs. Violin plots show Kernel density estimate KDE in green, interquartile ranges (25th/75th percentiles) in grey (squares), and the median in white (line). *P* values were obtained from GTEx, calculations are detailed in Oliva et al.[65]. **c** *CD2* expression in synovia from RA patients plotted against disease activity (DAS28-CRP). Data was extracted from GEO Dataset GSE45867. R[2] and *P* were calculated using simple liner regression. **d** Expression of *CD2* in synovial tissue from RA patients, osteoarthritis (OA) patients or healthy controls (GEO GDS5401-3). Females are shown in red and males in blue. **e** Expression of *CD2* in PBMCs from healthy males and females (GEO GDS5363). **f** CD2 expression on antigen experienced CD45RO[+] or naïve CD45RA[+] CD4[+] T cells from blood of a healthy donor. Data is representative of *n* = 3 independent human biological replicates. The experiment was done twice with similar results. **g**, **h** CD2 expression in CD45RO[+] T cells after 24 h incubation with 10–100 nM E2. Representative flow cytometry histogram showing CD2 expression in **g** and quantification in **h**. Data shown is pooled from two independent experiments. In all figures, each dot indicates one independent human biological replicate. Data are summarised as mean (SEM). In **d**, **e**, and **g**, statistical significance was evaluated using a two-tailed non-parametric Mann–Whitney U test. Sequential flow cytometry gating strategies for **f**–**h** are provided in Supplementary fig. 16.

immune response, both in homeostasis and in autoimmune conditions.

Sex-dependent differences in CD2 expression have implications for several sexually dimorphic immune processes involving T cells or other CD2 expressing cells. Hormonal regulation of CD2 could contribute to more vigorous humoral immune responses in women[2], helping to protect their off-spring from infections[52] at the cost of an enhanced risk to autoimmunity postpartum[53]. Alternatively, an enhanced CD2 expression in women might facilitate the induction of regulatory T cell phenotypes (as we observed in mice) to facilitate foetal-maternal immune tolerance. A hormonal regulation of CD2 expression could have wide ranging implications for personalized medicine in T-cell-mediated inflammatory diseases, as Alefacept was shown to preferentially target CD2[hi] T cells[31]. Indeed, we demonstrate strong effects of anti-Cd2 mAb administration on the naïve/effector T cell ratio in female, but not male mice. As such, sex-specific differences in T cell CD2 expression may offer a useful biomarker for stratification of patients in the context of CD2-targeted therapies.

In conclusion, our results highlight the importance of genotype-sex interactions for the sexual dimorphism in autoimmunity, demonstrating that sex can determine the penetrance of predisposing genetic factors. Our findings also show that CD2 is a sex-sensitive regulator of T-cell-mediated autoimmunity. Hormonal-mediated regulation of CD2 is a conserved mechanism that has implications for the sexual dimorphism in the susceptibility to -and treatment of- autoimmune diseases like RA.

## Methods

**Mice.** The BR.Cia21.D3-31 congenic founder mice were obtained from a partial advanced intercross (PAI) described elsewhere and they were subsequently back crossed for four additional generations[9]. In order to ensure strain purity, BR.Cia21.D3-31 mice were screened with a custom designed 8k Illumina chip at genome wide level[54] and the mice were found to be devoid of any contaminating RIIIS/J alleles. No SNPs were present between the congenic and the B10.RIII background strain. Mice were kept under specific pathogen free (SPF) conditions following FELASA II guidelines. Mice were housed in individually ventilated cages containing wood shavings in a climate-controlled environment (21–23 °C, 40–50% humidity) with a 12-h light-dark cycle, fed with standard chow and water ad libitum. All the experiments were performed with age-, sex- and cage-matched mice and all the genetic experiments were performed with littermate controls. All the experimental procedures were approved by the regional ethics committee Jordbruksverket in Stockholm, Swede. Ethical permit numbers; 12923/18 and N134/13 (genotyping and serotyping), N35/16 (CIA) and N83/13 (EAE).

**Preparation of mouse single cell suspensions.** Briefly, spleen or lymph nodes were harvested and mechanically dissociated on a 40 μM cell strainer (Falcon) using a 1 ml syringe plunger (Codan). Cells were counted on a Sysmex KX-21 cell counter. All centrifugation steps throughout the study were carried out a 350×*g* for 5 min at RT. For spleen samples, red blood cells were lysed in RBC buffer (155 mM NH4Cl, 12 mM NaHCO3, 0.1 mM EDTA) before counting.

**Preparation of human peripheral blood mononuclear cells.** Human peripheral blood mononuclear cells (PBMCs) were prepared from 8 ml whole blood of healthy donors (aged 28–40) using SepMate (Stemcell Technologies) tubes and Ficoll

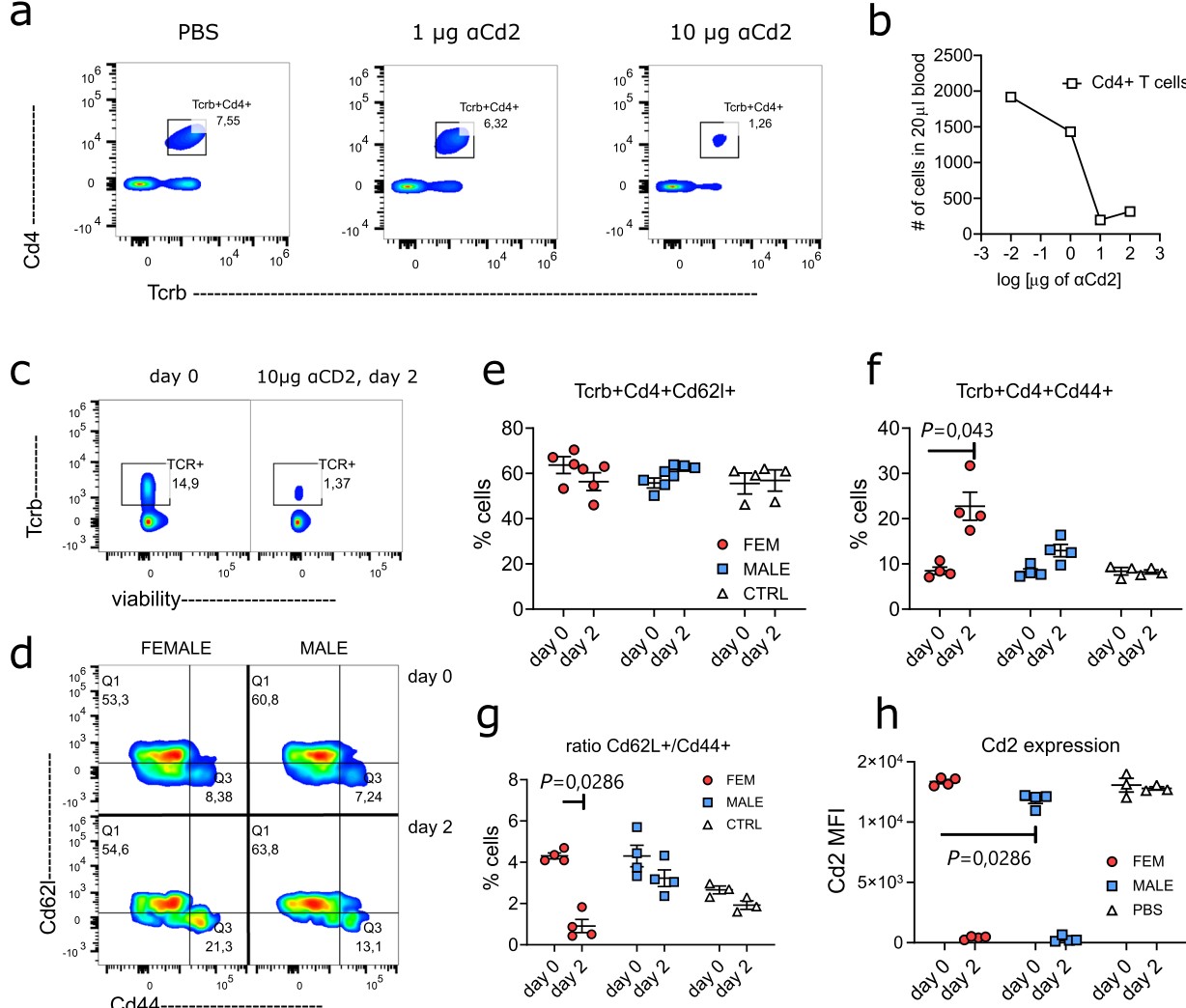

**Fig. 8 Anti-Cd2 is more effective at skewing T cell phenotypes in females. a**, **b** Initial titration experiment in mice showing in vivo depletion of T cells in dependency of administered anti-Cd2 mAb RM2-5. Representative flow cytometry plots of circulating blood T cells in **a** and total number of cells in **b**. Data shown is from one mouse. The experiment was performed three independent times with similar results. **c–h** Naïve BR male and female mice were injected i.p. with 10 μg anti-Cd2 mAb RM2-5. Circulating Cd4⁺ T cells were analysed before (day 0) and after (day 2) mAb injection. Representative flow cytometry plots are shown in **b** and **d**, quantification of the results including ratio of naïve (Cd62l⁺) to effector (Cd44⁺) Cd4⁺ T cells are shown in **e–h**. The experiment was performed two independent times with similar results. Individual dots represent independent mouse biological replicates. Data are summarised as mean (SEM). Statistical significance was evaluated using a two-tailed non-parametric Mann–Whitney *U* test. Sequential flow cytometry gating strategies for **a**–**h** are provided in Supplementary figs. 17–18.

density gradient medium (Sigma) according to the manufacturer. Ethical approval by the Swedish ethical Review authority Etikprövningsmyndigheten, Uppsala, Sweden. Ethical permit number: Dnr 2020-05001. Informed consent was received from all participants.

**Cell culture**. $10^6$ splenocytes, $5 \times 10^5$ lymph node cells, or $10^5$ PBMCs were cultured in 200 μl of complete RPMI per well in Nunclon U-shaped bottom 96-well plates (Thermo Scientific). Cells were incubated at 37 °C and 5% $CO_2$. Complete RPMI: RPMI 1640 with GlutaMAX™ (Thermo Scientific); 10% heat inactivated FBS (Thermo Scientific); 10 μM HEPES (Sigma); 50 μg/ml streptomycin sulfate (Sigma); 60 μg/ml penicillin C (Sigma); and 50 μM β-Mercaptoethanol (Thermo Scientific). FBS was heat-inactivated for 30 min at 56 °C. To assess the effect of 17-β-estradiol (Sigma) on CD2 expression, FBS was replaced with charcoal-stripped FBS (Thermo Scientific), and RPMI was replaced with RPMI without phenol red (Thermo Scientific). 17-β-estradiol was dissolved in ethanol.

**ELISA**. $10^6$ lymph node cells from CIA mice were plated per well and stimulated with 100 μg/ml bovine collagen type II (bCII) in complete RPMI for 48 h as described in cell culture. Supernatants were used for cytokine analysis. Flat 96-well plates (Maxisorp, Nunc) were coated overnight at 4 °C with the capture antibody (Ab, listed below) in PBS. After removing the coating solution, supernatant from

cell cultures were added. Plates were incubated for 3 h at RT before washing (0.05% Tween PBS) and adding the biotinylated detection Ab (listed below) in PBS (1 h at RT). Plates were washed and incubated 30 min at RT with Eu-labelled streptavidin (PerkinElmer, 1:1000) in 50 mM Tris-HCl, 0.9% (w/v) NaCl, 0.5% (w/v) BSA and 0.1% Tween 20, 20 μM EDTA. After washing, DELFIA Enhancement Solution (PerkinElmer) was added, and fluorescence read at 620 nm (Synergy 2, BioTek). Monoclonal antibodies (mAbs) to Il-2 (capture Ab 5 μg/ml JES6-IA12; detection Ab 2 μg/ml biotinylated-JES6-5H4, Mabtech), Il-17a (capture Ab 5 μg/ml TC11-18H10.1; detection Ab 2,5 μg/ml TC11-8H4, BD), Ifn-γ (capture Ab 5 μg/ml AN18; detection Ab 2,5 μg/ml biotinylated R46A2, Mabtech).

**Analysis of mRNA expression**. $10^6$ lymph node cells per well were stimulated for 24 h using mAb LEAF hamster anti-mouse Cd3 (1 μg/ml, 500A2, BD Pharmingen) and LEAF hamster anti-mouse Cd28 (1 μg/ml, 37.51, BD Pharmingen) as described in cell culture. Cells were washed in PBS and RNA was extracted using Qiagen RNeasy columns according to the manufacturer without DNAse digestion. RNA concentration was determined using a NanoDrop 2000 (Thermo Scientific). Sample concentrations were normalised before proceeding with reverse transcription. Samples were stored at −20 °C for short-term storage. cDNA synthesis was carried out using the iScript cDNA synthesis kit (Bio-Rad) according to the manufacturer. qRT-PCR primers covered an exon–exon junction to minimise amplification of genomic DNA and were used at a final concentration of 300 nM.

The qPCR reaction was carried out using the iQSYBR Green Mix (Bio-Rad) in white 96-well plates (Bio-Rad) using a CFX96 real-time PCR detection system (Bio-Rad). *Actb* or *Gapdh* were used as an internal control. Primer sequences are listed in Supplementary table 2. Data were analysed according to the ΔΔCt method[55], assuming equal efficiency for all the primer pairs.

**ChIP-qPCR**. $10 \times 10^6$ spleen cells/ml were fixed for 10 min in 1% formaldehyde PBS at RT. The reaction was stopped by adding 125 mM glycine and cells were washed twice in ice-cold PBS. Complete protease inhibitor cocktail (Roche) was added in all the following steps. $2 \times 10^6$ cells were lysed in 1 ml cell lysis buffer[56] on ice for 15 min, and the extracted nuclei lysed in 1 ml nuclear lysis buffer[56] on ice for 15 min. Lysates were sonicated for 15 cycles (on high settings, 30″ON-30″OFF) using a Diagenode Bioruptor. The water bath was cooled to 4 °C before beginning sonication. Average DNA length after sonication was 500 bp. 450 μl of the lysates were incubated with 10 μg/ml rabbit anti-mouse Erα Ab (clone E115, Abcam) or polyclonal rabbit IgG isotype control (Abcam) on a shaker at 4 °C overnight. Next day, DNA-Ab complexes were precipitated using protein G magnetic beads (Thermos Scientific). Beads were washed twice for 5 min at RT in buffers of increasing salt concentration according to Tantin et al.[56]. DNA was eluted by incubating beads in 100 μl elution buffer[56] at 65 °C for 30 min with occasional vortex. Beads were pelleted and fixation was reversed by incubation of supernatants for 8 h at 65 °C in the presence of 0.3 M NaCl in 96-well plates. On the third day, 10 μg/ml RNASe A (Thermo Scientific) was added for 30 min (37 °C) before incubation with 10 μg/ml of Proteinase K (Thermo Scientific) at 55 °C for 30 min. DNA was purified using GeneJET PCR purification kit (Thermo Scientific) and used for qPCR. Primers used for amplification of recovered DNA are listed in Supplementary table 3. Data was analysed according to[56], but briefly, results are presented as fold change over their respective mock IP controls.

**Flow cytometry**. $10^6$ cells were blocked in 20 μl of PBS containing 5 μg in-house produced 2.4G2 in 96-well plates for 10 min at RT. Samples were washed with 150 μl of PBS and subsequently stained with the indicated antibodies in 20 μl of PBS diluted 1:100 or 1:200 at 4 °C for 20 min in the dark (Ab list follows). Cells were washed once, fixed and permeabilized for intracellular staining using BD Cytofix/Cytoperm™ (BD) according to the manufacturer. Cells were stained intracellularly with 20 μl of permeabilization buffer (BD), using the antibodies at a 1:100 final dilution, for 20 min at RT. Foxp3 staining required nuclear permeabilization and was carried out using Bioscience™ Foxp3/Transcription Factor Staining Buffer. For intracellular cytokine staining, cells were stimulated in vitro with phorbol 12-myristate 13-acetate (PMA) 10 ng/ml, ionomycin 1 μg/ml, and BFA 10 μg/ml for 4–6 h at 37 °C prior to fixation, permeabilization and staining.

Flow cytometry anti-mouse antibodies (BD): Cd3 (clone: 145-2C11); Tcrb (H57-597); Cd4 (RM4-5); Cd8 (53-6.7); Cd19 (1D3, 6D5); Cd11b (M1/70); Cd11c (HL3, N418); Foxp3 (MF23); Cd25 (7D4); Cd44 (IM7); Cd62l (MEL-14); Cd2 (RM2-5); Ly6c (AL-21); Lag-3 (C9B7W); Cd40l (MR1); Ifn-γ (R46A2); and Il-17a (TC11-18H10.1). Cd16/Cd32 (2.4G2, in house).

Flow cytometry anti-human antibodies (BD): CD45 (clone: HI30); CD2 (RPA-2,10); TCRB (IP26); CD4 (OKT4); CD45RA (Hl100); and CD45RO (UCHL1).

**Proliferation assay**. $10^7$ lymph node cells were labelled using CellTrace™ Violet Cell Proliferation Kit (ThermoFisher Scientific) according to the manufacturer. $5 \times 10^5$ naïve lymph node cells were cultured per well in U 96-well plates as described under *cell culture* in the presence of hamster anti-mouse Cd3 (1 μg/ml, 500A2, BD Pharmingen) and hamster anti-mouse Cd28 (1 μg/ml, 37.51, BD) for 72–96 h. Proliferation by dilution of CTV was assessed using flow cytometry. Complementary antibody staining was done as described under flow cytometry. Proliferation parameters were analysed and calculated using FlowJo 8.8.7.

**Collagen-induced arthritis**. 12-week-old mice were immunised with 100 μg of bovine collagen type II (bCII) in 100 μl of a 1:1 emulsion with CFA (BD) and PBS intradermally at the base of the tail. Mice were challenged at day 35 with 50 μg of bCII in 50 μl of IFA (BD) emulsion. Mice were monitored for arthritis development as described in[57]. In short, each visibly inflamed (i.e. swollen and red) ankle or wrist was given 5 points, whereas each inflamed knuckle and toe joint was given 1 point each, resulting in a total of 60 possible points per mouse and day.

**Collagen antibody-induced arthritis**. CII-specific antibodies (M2139, CIIC1, CIIC2 and UL1) were generated and purified as previously described[15]. The sterile cocktail of M2139, CIIC1, CIIC2 and UL1 mAbs (4 mg per mouse) was injected intravenously. On day 7, lipopolysaccharide (O55:B5 LPS from Merck; 25 μg in 200 μl per mouse) was injected intraperitoneally to all mice to increase severity of the disease. Mice were scored as described for CIA.

**Experimental autoimmune encephalomyelitis**. 12-week-old mice were immunised with a 100 μl emulsion of 250 μg myelin basic protein peptide (MBP) 89-101 peptide in PBS and 50 μl IFA (incomplete Freud's adjuvant) containing 50 μg *Mycobacterium tuberculosis* H37RA (BD). Animals were boosted with 200 ng of *Bordetella pertussis* toxin (Sigma Aldrich, St. Louis, MO, USA) i.p. on day 0 and 48 h post initial immunisation. EAE severity was evaluated as described in[58]. Briefly, mice were scored as follows: 0, no clinical signs of disease; 1, tail weakness; 2, tail paralysis; 3, tail paralysis and mild waddle; 4, tail paralysis and severe waddle; 5, tail paralysis and paralysis of one limb; 6, tail paralysis and paralysis of two limbs; 7, tetraparesis; 8, moribund or deceased.

**Delayed type hypersensitivity**. A DTH reaction was elicited in the ear by immunising mice with 100 μg bCII (as described for CIA), and challenging 10 days later with 10 μg bCII intradermally in the dorsal part of the ear. The right ear was challenged with bCII, and the left one with vehicle (10 mM acetic acid). Ear swelling was measured at baseline, 48 h, and 72 h after challenge using a calliper. Plots show ear swelling as difference (Δ) to baseline.

**Ovariectomy**. In brief, ovaries of female mice were removed after a single incision through the back skin and bilateral flank incision through the peritoneum. Sham animals underwent the same procedure (i.e. incision in the back and peritoneum) without removing the ovaries. After the operation, mice were rested for a minimum of 14 days prior to immunisation for EAE or CIA as described elsewhere.

**Luciferase reporter assay**. $2 \times 10^4$ MCF-7 cells were seeded into flat 96-well flat bottom plates (Thermo Scientific) and left to adhere overnight. Then cells were transfected with pGL4.17 (Promega) luciferase reporter construct containing the BR or R3 allele of the candidate ERBS (pGL4.17.BR and pGL4.17.R3, respectively). ERBS cloning primers 5′-3′, Fw: AGATCTCGAGGGGGAAAGCTCTGACTT GGG; Rv: GTCAAGCTTGAGAAAGAATTTTGCTTATTTAGTCC. Cells were transfected in OPTIMEM medium (Thermo Scientific) using lipofectamine 3000 (Thermo Scientific) according to the manufacturer. The transfection mix (per well) contained 400 ng plasmid, 0.3 μl lipofectamine and 0.2 μl P3000 reagent. Respective stimuli (20 ng/ml PMA, 10-100 nM E2) were added after 24 h, and cells were further incubated overnight before lysis. Luciferase activity was measured using Pierce Firefly Luc One-Step Glow Assay Kit (Thermo Scientific) in a Synergy-2 plate reader (BioTek).

**SNP sequencing and sequence comparison**. SNP sequencing was performed on a Qiagen PSQ HS 96 Pyrosequencer using PyroMark Gold reagents (Qiagen) according to the manufacturer. Sequences were compared using Clustal Omega v1.2.4[59].

**Genetic association study**. Data for genetic variations within *CD2-CD58* locus was extracted from previous Immunochip data published elsewhere (PMID: 23143596). After filtering these data correspond to 263 SNPs in 1940 healthy controls (M/F 524/1416) and 2762 RA patients (M/F 817/1945) from the Swedish EIRA study. Association was analysed by PLINK separately for female and male individuals. Data from IEU Open GWAS database used in Supplementary fig. 8 was analysed using Ieugwasr[60] in R v4.1.

**Analysis of public microarray expression data**. Microarray data was extracted from NCBI GEO Database[29] and analysed using GEO2R (R v3.2.3; Biobase v2.30.0; GEOquery v2.40.0; limma v3.26.8) as well as Shiny GEO[61]. GEO accession numbers are mentioned in figure legends and the data availability statement.

**Analysis of public ChIP-seq data**. Data was obtained from ChIP-Atlas database[30] and visualised using IGV v2.9.4. Accession numbers are mentioned in figure legends and the data availability statement.

**Statistical analysis**. Statistical analysis was performed using GraphPad Prism v6.0 or higher. Statistical significance was evaluated using non-parametrical two-tailed Mann–Whitney *U* test unless stated otherwise in the figure legends. *P* values under 0.05 were considered statistically significant.

**Proteomic analysis of enriched Cd4+ T cells**. Cd4+ T cells were enriched from naïve spleens using untouched Cd4+ T cell mouse kit (Dynabeads, Life Technologies, purity 82%). 96-well U bottom plates were pre-coated with 1 μg/ml of anti-Cd3 and 1 μg/ml of anti-Cd2 in PBS for 3 h at 37 °C. $2.5 \times 10^5$ Cd4+ T cells were plated on the pre-coated plates and cultured for 48 h.

Cell pellets were lysed in a buffer consisting of 1% SDS, 8 M urea and 20 mM EPPS pH 8.5 and sonicated using a Branson probe sonicator (3 s on, 3 s off pulses, 45 s, 30% amplitude). Protein concentration was measured using BCA assay and subsequently 50 μg of protein from each sample were reduced with 5 mM DTT at RT for 45 min followed by alkylation with 15 mM IAA in the dark at RT for 45 min. The reaction was quenched by adding 10 mM DTT and the samples were precipitated using methanol-chloroform mixture. Dried protein pellets were dissolved into 8 M urea, 20 mM EPPS pH 8.5. EPPS (20 mM, pH 8.5) was added to lower the urea concentration to 4 M and LysC digestion was done at a 1:100 ratio (LysC/protein, w/w) overnight at RT. Then urea concentration was lowered to 1 M and trypsin digestion was conducted at a 1:100 ratio (Trypsin/protein, w/w) at RT for 5 h. TMTpro plex (Thermo Fischer Scientific) reagents were dissolved into dry

acetonitrile (ACN) to a concentration of 20 μg/μl and 200 μg were added to each sample. The ACN concentration in the samples was adjusted to 20% and the labelling was conducted at RT for 2 h and quenched with 0.5% hydroxylamine (ThermoFischer Scientific) for 15 min at RT. The samples were then combined and dried using Speedvac to eliminate the ACN. Then samples were acidified to pH < 3 using TFA and desalted with SepPack (Waters). Lastly, peptide samples were dissolved into 20 mM NH4OH and 150 μg of each sample was used for off-line fractionation.

Samples were fractionated off-line in a high-pH reversed-phase manner using an UltimateTM 3000 RSLCnano System (Dionex) equipped with a XBridge Peptide BEH 25 cm column of 2.1 mm internal diameter, packed with 3.5 μm C18 beads having 300 Å pores (Waters). The mobile phase consisted of buffer A (20 mM $NH_4OH$) and buffer B (100% ACN). The gradient started from 1% B to 23.5% B in 42 min, then to 54% B in 9 min, 63% B in 2 min and stayed at 63% B for 5 min and finally back to 1% B and stayed at 1% B for 7 min. This resulted in 96 fractions that were concatenated into 24 fractions. Samples were then dried using Speedvac and re-suspended into 2% ACN and 0.1% FA prior to LC-MS/MS analysis.

Peptides were separated on a 50-cm EASY-spray column, with a 75 μm internal diameter, packed with 2 μm PepMap C18 beads, having 100 Å pores (Thermo Fischer Scientific). An UltiMate™ 3000 RSLCnano System (Thermo Fischer Scientific) was used that was programmed to a 91 min optimised LC gradient. The two mobile phases consisted of buffer A (98% milliQ water, 2% ACN and 0.1% FA) and buffer B (98% ACN, 2% milliQ water and 0.1% FA). The gradient was started with 4% B for 5 min and increased to 26% B in 91 min, 95% B in 9 min, stayed at 95% B for 4 min and finally decreased to 4% B in 3 min and stayed at 4% B for 8 more min. The injection was set to 5 μL corresponding to approximately 1 μg of peptides.

Mass spectra were acquired on a Q Exactive HF mass spectrometer (Thermo Fischer Scientific). The Q Exactive HF acquisition was performed in a data dependent manner with automatic switching between MS and MS/MS modes using a top-17 method. MS spectra were acquired at a resolution of 120,000 with a target value of $3 \times 10^6$ or maximum integration time of 100 ms. The $m/z$ range was from 375 to 1500. Peptide fragmentation was performed using higher-energy collision dissociation (HCD), and the normalised collision energy was set at 33. The MS/MS spectra were acquired at a resolution of 60,000 with the target value of $2 \times 10^5$ ions and a maximum integration time of 120 ms. The isolation window and first fixed mass were set at 1.6 $m/z$ units and $m/z$ 100, respectively.

**TMT-10 labelling quantification.** Protein identification and quantification were performed with MaxQuant software (version 1.6.2.3). MS2 was selected as the quantification mode and masses of TMTpro labels were added manually and selected as peptide modification. Acetylation of N-terminal, oxidation of methionine and deamidation of asparagine and glutamine were selected as variable modifications while carbamidomethylation of the cysteine was selected as fixed modification. The Andromeda search engine was using the UP000000589_Mus musculus database (22129 entries) with the precursor mass tolerance for the first searches and the main search set to 20 and 4.5 ppm, respectively. Trypsin was selected as the enzyme, with up to two missed cleavages allowed; the peptide minimal length was set to seven amino acid. Default parameters were used for the instrument settings. The FDR was set to 0.01 for peptides and proteins. "Match between runs" option was selected with a time window of 0.7 min and an alignment time window of 20 min.

**Reporting summary.** Further information on research design is available in the Nature Research Reporting Summary linked to this article.

## Data availabilty

Mass spectrometry proteomics data have been deposited in the ProteomeXchange Consortium via the PRIDE partner repository[62] under the accession code PXD024126.

The source data for all other figures (i.e. Figs. 1, 2, 3, 4a, c–e, 5, 6, 7a, g and 8b, c) has been deposited in Figshare under accession code 14685906.

The accession codes and hyperlinks for publicly available data that we accessed are provided as follows:
Fig. 4c ChIP-Atlas "SRX129062".
Fig. 7b NCBI dbGaP "phs000424.v8.p2".
Fig. 7c: NCBI GEO "GSE45867"; Fig. 7d: NCBI GEO "GDS5401"; Fig. 7e: NCBI GEO "GDS5363"; Supplementary fig. 7: NCBI GEO "GSE5603"; Supplementary fig. 8: ChIP-Atlas "SRX1995230", ChIP-Atlas "SRX3447357", IEU Open GWAS "ukb-d-M13_RHEUMA", IEU Open GWAS "bbj-a-72", IEU Open GWAS "finn-a-M13_RHEUMA", IEU Open GWAS "ieu-a-832", IEU Open GWAS "ebi-a-GCST005569" and IEU Open GWAS "ebi-a-GCST000679".

Source data are provided with this paper as a Source Data file. Source data are provided with this paper.

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

## Acknowledgements

We would like to thank Dr Leonid Padyukov and the EIRA study group for providing the genetic data. This work was supported by grants from the Knut and Alice Wallenberg Foundation, the Swedish Association against Rheumatism, the Swedish Medical Research Council, the Swedish Foundation for Strategic Research and Karolinska Institute-KID.

## Author contributions

G.F.L. wrote the manuscript with the help of M.F. and R.H.; G.F.L. designed, performed, analysed and interpreted most experiments. M.F. and M.J. designed, performed and analysed all the experiments shown in Figs. 1b, 2 and 3. R.Z. and P.S. performed and analysed experiments requiring mass spectrometry (Fig. 6h). K.S.N. helped secure funding and reviewed the manuscript. E.L., M.A. and Y.H. helped with data collection, analysis and interpretation. All authors revised and approved the manuscript. R.H. initiated, designed and supervised the project and takes overall responsibility for the data.

## Funding

## Competing interests

The authors declare no competing interests.
