## [Peer Review File · Nature Communications]

REVIEWER COMMENTS

Reviewer #1 (Remarks to the Author):

This is an important paper that investigates a variant Cia21 QTL that imparts resistance to mouse autoimmune models that typically have a female bias. They map the QTL to a region containing the adhesion molecule CD2. Female seems to have naturally high CD2 expression and the variant increases estrogen receptor binding and lowers CD2 expression. The increased CD2 expression is related to increase IL-17 production in the mouse models and humans show a similar binding site in the vicinity of CD2 with a similar regulation pattern with high CD2 expression correlating with disease score in RA. I have a few questions.

1. Males have the same level of CD2 expression as females with the protective variant. What causes females with the BR variant to have higher CD2 expression than males? Is there a different repression mechanism in males that keeps CD2 expression relatively low when the ER binding in females is weaker?

2. I don't fully understand the discussion of LAG3 expression increasing with anti-CD2. Is the application of CD2 in this case agonistic as in activating anti-CD2 on a surface, or is it inhibitory, such as adding anti-CD2 to block CD2-CD58 mediated adhesion. If agonistic, they show that activation through CD2 increases LAG3 expression. Wouldn't this make having more CD2 protective against autoimmunity as LAG3 is an inhibitory receptor. I the authors must need to clarify when they are presenting anti-CD2 as a surface bound costimulation vs adding it to a system in solution to block adhesion/co-stimulation.

3. It appears that anti-CD2 antibodies down-regulation CD2 expression to near zero and may cause an imbalance in the ratio of T cell subsets. Alefacept depletes effector memory T cells, so was seen to increase (naïve + central memory) / effector memory ratio. I don't understand why this ratio goes the opposite direction in female mice treated with anti-CD2 mAb. In this case, the CD2 antibody strongly down-regulates CD2 expression and appears to deplete naïve and central memory cells. In addition to the ratios, can the authors provide absolute cell counts to support that depletion is taking place? The other possibility is that the anti-CD2 antibody is activating cells and shifting some cells to CD44 high and this happening more in the females due to higher initial CD2 expression. It would be helpful to clarify this.

Reviewer #2 (Remarks to the Author):

The authors have presented interesting findings that have implications for how estrogen hormones may regulate the immune response. The results have implications in terms of the pathogenesis of autoimmune diseases such as rheumatoid arthritis. Studies presented here use forward genetics to build on their previous work investigating hormone-immune system interactions to now develop further insights into how T cell activation might be triggered by estrogen. The studies clearly demonstrate that the T cell co-stimulatory molecule CD2 is regulated by estradiol. An overall strength is that the findings are demonstrated in both mouse models of disease and in human samples. Specific strengths as well as items for further consideration or clarification follow:

1. Data in the mouse models are strongly supportive of the sex-specific responses; animals have been appropriately manipulated with ovariectomy. Missing from the methods (line 449) is description of the sham ovx procedure.

2. Estrogen responses are in the low nM range (10 nM) which is physiologic. The cultures for the in vitro studies were carried out with charcoal-stripped FCS which removes added estrogen and other hormones. However, there is no mention of whether the medium contained Phenol Red, which has weak estrogenic effects and could obscure some responses at lower hormone levels.

3. The diminished T cell response in D3-31 mice with CIA is shown by reduced production of cytokines. However, the finding that numbers of induced T regs after immunization were lower in females seems to go in the opposite direction, predictive of less suppression of the immune response (Figure 6); the discussion notes that Tregs were induced (line 305); this is somewhat confusing and further clarification would be useful.

5. Demonstration of higher expression of CD2 in RA synovium, compared to healthy and OA groups, may reflect the greater numbers of infiltrating T cells, rather than any specificity for the RA disease process (Figure 7). Differences in CD2 expression in females and males (7e) do not appear to be statistically significant; the finding is suggestive but not definitive (line 294). It is somewhat surprising that the CIA mice were not examined for synovial CD2 expression.

6. The age ranges of healthy human donors are not provided (line 341); since pre-menopausal females would be most relevant to these studies, age is a relevant variable. It would be interesting to know whether CD2 expression is lower in women without endogenous estrogens, after menopause.

Other minor points:

1. In the introduction, lines 39-41, the section starting "Not only through sex chromosomes..." is a fragment of a sentence.
2. Line 351: "solved" should be "dissolved."
3. Line 363: the source should be "Biolegend."

Reviewer #3 (Remarks to the Author):

This work examines the role of a sexually dimorphic locus in mice that regulates resistance to a T dependent form of collagen induced arthritis in mice. The authors do an elegant job of mapping and isolating at least one of the functional elements in the CIA21 locus, and convincingly demonstrate that this is related to polymorphisms in an estrogen response element that controls expression of CD2. Interestingly, female and estrogen effects at this locus repress CD2, thereby preventing its negative regulatory effect on T cell responses, perhaps in part via the actions of LAG-3.

The authors go on to explore the sex dependent effect of variation at this locus in humans. Here the data is large correlative. The associations with RA are modest, as are the effects of female sex on expression of CD2. Nevertheless, the data does support a relationship between CD2 and disease activity in RA. The data would be strengthened by a more detailed analysis of the CD2 locus in humans in order to demonstrate directly that an estrogen dependent mechanisms might explain the genetic associations in humans. Of course, the estrogen effect could be independent of genetics at the CD2 locus and still be relevant to sex effects on disease risk. But the authors imply that the genetic effects are dependent on sex which is not entirely clear.

Overall, these data continue to support the generally accepted hypothesis that estrogen mediated regulation of the immune response and may account for some of bias towards female predominance of autoimmune disorders.

We are grateful for the interest shown in our findings and the time and efforts spent by the reviewers to give thoughtful suggestions in order to improve our manuscript. We have addressed all questions and comments, and have included new data where needed. Changes made to the original manuscript are highlighted in the revised version (yellow). Fig. 7 has been split into figs. 7 and 8 in the new version to improve the comprehensibility of the manuscript. Our point-by-point response to the reviewers follows below (in blue).

REVIEWER COMMENTS

Reviewer #1 (Remarks to the Author):

This is an important paper that investigates a variant Cia21 QTL that imparts resistance to mouse autoimmune models that typically have a female bias. They map the QTL to a region containing the adhesion molecule CD2. Female seems to have naturally high CD2 expression and the variant increases estrogen receptor binding and lowers CD2 expression. The increased CD2 expression is related to increase IL-17 production in the mouse models and humans show a similar binding site in the vicinity of CD2 with a similar regulation pattern with high CD2 expression correlating with disease score in RA. I have a few questions.

1. Males have the same level of CD2 expression as females with the protective variant. What causes females with the BR variant to have higher CD2 expression than males? Is there a different repression mechanism in males that keeps CD2 expression relatively low when the ER binding in females is weaker?

Authors: We propose that estradiol enhances CD2 expression in female wildtype BR mice but fails to do so in female congenic D3-31. Likely, this is due to the identified variant affecting binding of activated ER to the DNA (fig. 4e). Thus, CD2 in males is expressed at 'baseline' levels, and estrogen acts as a potentiator. The notion of estrogen as an enhancer of CD2 expression is supported by fig. 5f (in mice) and fig. 7 (in humans).

2. I don't fully understand the discussion of LAG3 expression increasing with anti-CD2. Is the application of CD2 in this case agonistic as in activating anti-CD2 on a surface, or is it inhibitory, such as adding anti-CD2 to block CD2-CD58 mediated adhesion. If agonistic, they show that activation through CD2 increases LAG3 expression.

Authors: As the reviewer points out, our data shows that anti-CD2 RM2-5 (the clone used in this study) inhibits T cell activation while it promotes LAG-3 expression. The literature (1), together with our data, suggest that RM2-5 blocks CD2-CD48 interactions in vitro. CD2-CD48 interaction in mice, as well as CD2-CD58 interaction in humans, are important not only for T cell-APC, but also T cell-T cell interactions, as activated T cells potently express CD48/CD58 (supplementary fig. S3 and (2)). Costimulation through CD2 enhances T cell activation, and we propose that inadequate CD2 costimulation results in upregulation of inhibitory or anergic markers such as LAG-3. We have now formulated this more clearly in the Abstract, Results, and Discussion (highlighted in yellow).

Wouldn't this make having more CD2 protective against autoimmunity as LAG3 is an inhibitory receptor.

Authors: We agree that this could seem initially confusing, but as explained above, we reason that CD2-mediated costimulation primarily enhances T cell activation.

I the authors must need to clarify when they are presenting anti-CD2 as a surface bound costimulation vs adding it to a system in solution to block adhesion/co-stimulation.

Authors: We have now added this information to the figure legends in in figs. 5 and 6. We used soluble RM2-5 in all experiments except for the proteomic data in fig. 6h, where we used plate bound RM2-5. Please note that plate bound RM2-5 also blocks T cell activation (Supplementary fig. S4).

3. It appears that anti-CD2 antibodies down-regulation CD2 expression to near zero and may cause an imbalance in the ratio of T cell subsets. Alefacept depletes effector memory T cells, so was seen to increase (naïve + central memory) / effector memory ratio. I don't understand why this ratio goes the opposite direction in female mice treated with anti-CD2 mAb.

Authors: Alefacept and RM2-5 target different epitopes on the CD2 molecule, resulting in different downstream effects. It has been shown that anti-CD2 monoclonal antibodies targeting different epitopes diverge in their capacity to block adhesion or stimulate proliferation (3).

In this case, the CD2 antibody strongly down-regulates CD2 expression

Authors: Please note that RM2-5 does not downregulate CD2 expression but instead blocks the detection antibody (same clone) used for flow cytometry.

and appears to deplete naïve and central memory cells. In addition to the ratios, can the authors provide absolute cell counts to support that depletion is taking place? The other possibility is that the anti-CD2 antibody is activating cells and shifting some cells to CD44 high and this happening more in the females due to higher initial CD2 expression. It would be helpful to clarify this.

Authors: Note that fig. 7 has now been split into figs. 7 and 8 to improve comprehensibility. RM2-5 clearly depletes T cells as we now show in fig. 8a and b. However, depletion is only partial, and CD44+ cells are overrepresented in the remaining T cell pool (fig. 8b and c). RM2-5 might activate, rather than delete, a subset of T cells (as the reviewer suggests). This shift toward CD44+ is enhanced in females, probably due to higher initial CD2 expression.

Reviewer #2 (Remarks to the Author):

The authors have presented interesting findings that have implications for how estrogen hormones may regulate the immune response. The results have implications in terms of the pathogenesis of autoimmune diseases such as rheumatoid arthritis. Studies presented here use forward genetics to build on their previous work investigating hormone-immune system interactions to now develop further insights into how T cell activation might be triggered by estrogen. The studies clearly demonstrate that the T cell co-stimulatory molecule CD28 is regulated by estradiol. An overall strength is that the findings are demonstrated in both mouse models of disease and in human samples. Specific strengths as well as items for further consideration or clarification follow:

1. Data in the mouse models are strongly supportive of the sex-specific responses; animals have been appropriately manipulated with ovariectomy. Missing from the methods (line 449) is description of the sham ovx procedure.

Authors: The details on the sham ovx procedure have now been added.

2. Estrogen responses are in the low nM range (10 nM) which is physiologic. The cultures for the in vitro studies were carried out with charcoal-stripped FCS which removes added estrogen and other hormones. However, there is no mention of whether the medium contained Phenol Red, which has weak estrogenic effects and could obscure some responses at lower hormone levels.

Authors: As the reviewer correctly point out, phenol red is known to have weak estrogenic effects. For this reason, phenol red was omitted from cultures when testing estrogenic effects in vitro (i.e. fig. 4e, fig. 5f and g, fig. 6b, and fig. 7g). We have now specified this in the Methods section under *Cell culture*.

3. The diminished T cell response in D3-31 mice with CIA is shown by reduced production of cytokines. However, the finding that numbers of induced T regs after immunization were lower in females seems to go in the opposite direction, predictive of less suppression of the immune response (Figure 6);

Authors: The reviewer correctly points out that Treg frequencies in fig. 6 are seemingly opposite to what one would expect. However, increased Treg frequencies can be indicative of ongoing inflammation, as Treg frequencies are generally higher in immunized mice (suppl. fig. S5a). Further, Tregs are a heterogeneous population, and some cells could have only weak suppressive capacity (4). Similarly, FOXP3 can be an activation marker in T effector cells without conferring regulatory properties (5).

the discussion notes that Tregs were induced (line 305); this is somewhat confusing and further clarification would be useful.

Authors: Treg differences between D3-31 and BR mice were only evident in immunized mice, therefore we concluded that these cells must have been induced (suppl. fig. S5b).

5. Demonstration of higher expression of CD2 in RA synovium, compared to healthy and OA groups, may reflect the greater numbers of infiltrating T cells, rather than any specificity for the RA disease process (Figure 7).

Authors: We have now normalized expression of CD2 to expression of CD4 to approximate and account for differences in infiltrating T cells. As shown in suppl. fig. S6, CD2 is still overexpressed in RA synovia and in healthy females after normalizing for CD4 expression.

Differences in CD2 expression in females and males (7e) do not appear to be statistically significant; the finding is suggestive but not definitive (line 294).

Authors: The CD2 expression data shown in fig. 7e was calculated with the Mann Whitney test and was in fact found to be significant.

It is somewhat surprising that the CIA mice were not examined for synovial CD2 expression.

Authors: While this certainly would be interesting complementary data, we didn't consider it essential at the time of writing given that we already convincingly show sex-dependent differential expression of CD2 in T cells on an mRNA and on protein level (figs. 5 and 6).

6. The age ranges of healthy human donors are not provided (line 341); since pre-menopausal females would be most relevant to these studies, age is a relevant variable.

Authors: The age of the healthy donors has been added to the Methods sections. Samples were from volunteers aged 28-35.

It would be interesting to know whether CD2 expression is lower in women without endogenous estrogens, after menopause.

Authors: We analyzed CD2 expression in dependency of age in PBMCs from healthy women (suppl. fig. S7), but did not find significant differences. One should consider that there are important confounding factors, such as the use of hormone

replacement therapy to manage menopause symptoms, that might obscure the analysis.

Other minor points:

1. In the introduction, lines 39-41, the section starting “Not only through sex chromosomes...” is a fragment of a sentence.
2. Line 351: “solved” should be “dissolved.”
3. Line 363: the source should be “Biolegend.”

Authors: We have fixed all minor points.

Reviewer #3 (Remarks to the Author):

This work examines the role of a sexually dimorphic locus in mice that regulates resistance to a T dependent form of collagen induced arthritis in mice. The authors do an elegant job of mapping and isolating at least one of the functional elements in the CIA21 locus, and convincingly demonstrate that this is related to polymorphisms in an estrogen response element that controls expression of CD2. Interestingly, female and estrogen effects at this locus repress CD2, thereby preventing its negative regulatory effect on T cell responses, perhaps in part via the actions of LAG-3. The authors go on to explore the sex dependent effect of variation at this locus in humans. Here the data is large correlative.

The associations with RA are modest, as are the effects of female sex on expression of CD2. Nevertheless, the data does support a relationship between CD2 and disease activity in RA. The data would be strengthened by a more detailed analysis of the CD2 locus in humans in order to demonstrate directly that an estrogen dependent mechanisms might explain the genetic associations in humans.

Authors: We now include a small metanalysis more clearly demonstrating the association of the CD2 locus to RA. The data was extracted from the IEU open GWAS project (6), a curated database containing summary statistics of different GWAS datasets. Although we lack the possibility to stratify these data by gender (information is not provided), we have leveraged these data to investigate whether risk conferring polymorphisms in the CD2 locus coincide with ER binding sites. We find that this is the case (see supplementary fig. S8) but lack the possibility to test the relevance of these findings experimentally.

Of course, the estrogen effect could be independent of genetics at the CD2 locus and still be relevant to sex effects on disease risk. But the authors imply that the genetic effects are dependent on sex which is not entirely clear.

Authors: Certainly, genetics and sex related factors can play independent roles in disease susceptibility. However, we convincingly demonstrate in the mouse that genotype-sex interactions at the CD2 locus is one of the factors regulating the susceptibility to autoimmunity.

Overall, these data continue to support the generally accepted hypothesis that estrogen mediated regulation of the immune response and may account for some of bias towards female predominance of autoimmune disorders.

References

1. Y. Latchman, H. Reiser, *Eur. J. Immunol.* (1998), doi:10.1002/(sici)1521-4141(199812)28:12<4325::aid-immu4325>3.0.co;2-t.
2. A. Vandebon *et al.*, *Proc. Natl. Acad. Sci.* (2016), doi:10.1073/pnas.1604351113.
3. S. C. Meuer *et al.*, *Cell* (1984), doi:10.1016/0092-8674(84)90039-4.
4. X. Chen *et al.*, *J. Immunol.* (2008), doi:10.4049/jimmunol.180.10.6467.
5. S. E. Allan *et al.*, *Int. Immunol.* (2007), doi:10.1093/intimm/dxm014.
6. B. Elsworth *et al.*, The MRC IEU OpenGWAS data infrastructure. *bioRxiv* (2020), , doi:10.1101/2020.08.10.244293.
7. R. Edgar, M. Domrachev, A. E. Lash, *Nucleic Acids Res.* (2002).
8. D. A. Lury, R. A. Fisher, *Stat.* (1972), doi:10.2307/2986695.
9. Y. Kochi *et al.*, *Nat. Genet.* (2010), doi:10.1038/ng.583.
10. K. Ishigaki *et al.*, Large scale genome-wide association study in a Japanese population identified 45 novel susceptibility loci for 22 diseases. *bioRxiv* (2019), , doi:10.1101/795948.
11. Y. Okada *et al.*, *Nature* (2014), doi:10.1038/nature12873.
12. S. Eyre *et al.*, *Nat. Genet.* (2012), doi:10.1038/ng.2462.
13. E. A. Stahl *et al.*, *Nat. Genet.* (2010), doi:10.1038/ng.582.

REVIEWERS' COMMENTS

Reviewer #1 (Remarks to the Author):

The authors have addressed or clarified my concerns. I'm surprised that the solid phase anti-CD2 is inhibitory when co-immobilized on a surface with anti-CD3, but accept the authors observations regarding this. In the human system, the authors may want to use CD2.1 antibody clone for staining to avoid the epitopes being blocked by the antibody used for treatment. I'm not aware of such an antibody for mouse CD2. I have no further comments.

Reviewer #2 (Remarks to the Author):

All of my major questions and concerns have been addressed in the revision.

I have only one additional minor point. The caption for supplemental figure S7 might be corrected from "in dependency of age", to either "in relation to age" or " as a function of age" to clarify the usage.

Reviewer #3 (Remarks to the Author):

This manuscript is improved and communicates an important example of hormonal regulation of genes related to autoimmune arthritis susceptibility. The public human data support the model of susceptibility alleles mediating estrogen regulation of CD2, but of course definitive proof will require detailing this relationship in human samples.